# Hybrid Distillation: Connecting Masked Autoencoders with Contrastive Learners

## Abstract

Representation learning has been evolving from traditional supervised training to Contrastive Learning (CL) and Masked Image Modeling (MIM). Previous works have demonstrated their pros and cons in specific scenarios, *i.e.*, CL and supervised pre-training excel at capturing longer-range global patterns and enabling better feature discrimination, while MIM can introduce more local and diverse attention across all transformer layers. In this paper, we explore how to obtain a model that combines their strengths. We start by examining previous feature distillation and mask feature reconstruction methods and identify their limitations. We find that their increasing diversity mainly derives from the asymmetric designs, but these designs may in turn compromise the discrimination ability. In order to better obtain both discrimination and diversity, we propose a simple but effective Hybrid Distillation strategy, which utilizes both the supervised/CL teacher and the MIM teacher to jointly guide the student model. Hybrid Distill imitates the token relations of the MIM teacher to alleviate attention collapse, as well as distills the feature maps of the supervised/CL teacher to enable discrimination. Furthermore, a progressive redundant token masking strategy is also utilized to reduce the distilling costs and avoid falling into local optima. Experiment results prove that Hybrid Distill can achieve superior performance on different benchmarks.

## 1 Introduction

Pre-training followed by fine-tuning has been a common paradigm for computer vision tasks since the advent of deep learning. In the past decade, supervised image classification [16, 10, 24] over the widely used ImageNet [32] has dominated the pretraining mode. Recently, self-supervised learning has emerged as a promising alternative, particularly with two approaches: Contrastive Learning (CL) and Masked Image Modeling (MIM). The former one, typical representatives are MoCo [14] and SimCLR [4], learns invariant representation for positive views, which are usually defined as different augmentations of the same image. Furthermore, CLIP [30] extends CL to a multi-modal manner, which utilizes the corresponding text description of the given image as positive pairs. While the latter, including MAE [13] and SimMIM [44], aims to reconstruct the masked image patches and has become mainstream due to its efficiency brought by mask operations.

The different pre-training paradigms of CL and MIM facilitate a series of studies [43, 27, 38] that aim at understanding their respective properties. These studies point out that CL pre-training behaves more similar to supervised pre-training, *i.e.*, it provides models with longer-range global patterns targeting object shape, particularly in the last few layers [27], and enables feature representation with better **discrimination**. However, as shown in Fig. 1(a), CL pre-training causes self-attention in the last few layers to collapse into homogeneity, with attention distances located within a very small distance range. In contrast, MIM pre-training can bring more diverse attention and evenly distributed representations to all layers [43, 27], and this **diversity** contributes to its better generalization on

downstream fine-tuning. Nevertheless, MIM pre-training is slower to converge and underperforms in linear probing, mainly due to its lack of discrimination ability.

Since discrimination and diversity are both crucial for downstream adaptation, previous methods [41, 11, 23, 40, 29] propose to utilize feature distillation to combine the benefits of CL and MIM. Among them, dBOT [23] replaces the reconstructing objective of MAE with the feature maps of different pre-trained teachers. It finds that feature distillation can bring diverse attention no matter what the teacher model is, and the performance is comparable across different teachers, even with the randomly initialized ones, after multi-stage distillation. Also observing that distillation can yield diversity benefits, FD [41] directly distills feature maps from supervised/CL teachers to relieve the attention collapse and achieves considerable downstream performance gains. Although interesting and important, we argue that their findings are incomplete.

This paper re-examines these findings and reconsiders the importance of diversity and discrimination. Our study reveals the following observations: (i) **The increase in diversity derives from the asymmetric architecture designs, rather than feature distillation itself.** (Section 2.2) After removing the asymmetric attention in [41] and encoder-decoder designs in [23] and keeping the same teacher and student structures, we observe a negligible increase (or even a decrease) in attention diversity. (ii) **The asymmetric decoder de facto harm the discrimination over the encoder side, for it migrates the semantic information of the teacher model.** (Section 2.3) Due to the decomposition of the encoding and decoding functions, student encoders tend to summarize more general information, thus gradually losing the semantics obtained from teachers and yielding similar results after multi-stage distillation [23]. (iii) **Mask reconstruction of high-level semantics does not help improve diversity.** (Section 2.4) The phenomenon of reconstructing high-level information [29, 11, 40] is similar to direct feature distillation and lacks the diversity found in MIM, which implies that the attention diversity of MIM mainly comes from low-level reconstruction objectives.

Based on the above observations, we argue that a better distillation strategy is needed to help student models inherit both diversity and discrimination. To this end, we propose a simple but effective feature distillation method, termed as **Hybrid Distill**, to fully exploit the pre-trained model. Unlike previous works, Hybrid Distill aims to distill knowledge from both the supervised/CL and MIM teacher, allowing the student model to benefit from their respective advantages. To realize this, Hybrid Distill makes careful designs for the distilling target and location. Specifically, we find that **the relational modeling ability of MIM is crucial for preserving token diversity, while the feature maps of supervised/CL teachers are beneficial for discrimination**. Accordingly, we set the token relations of the MIM teacher and the feature maps of the supervised/CL teacher as the distilling objectives of Hybrid Distill. The token relations are distilled in layers preceding the final layer where attention collapse tends to occur, while the feature maps are distilled in the final layer to preserve semantics. Additionally, Hybrid Distill proposes a progressive redundant token masking strategy to reduce distilling costs and prevent falling into local optima. Experiment results show that the above distilling strategy works surprisingly well even when using MAE and CLIP teachers, *i.e.*, MAE pretrained with only 1.28M ImageNet images can also boost the large-scale (400M) pretrained CLIP teacher on different downstream tasks.

In a nutshell, this paper makes the following distribution:

• We re-examine the findings of previous feature distilling methods and point out that their increasing diversity mainly arises from the use of asymmetric designs, while these designs may in turn compromise the discrimination.

• We further propose a Hybrid Distill framework that utilized both supervised/CL and MIM teacher to provide the student with higher-quality discrimination and diversity. Distilling targets and locations are carefully designed in Hybrid Distill to fully exploit the strengths of both teachers.

• We conduct property analysis to demonstrate that the representations exhibit both discrimination and diversity in our Hybrid Distill. Experiments on various downstream tasks, including classification, detection, and segmentation, also showcase its superiority.

## 2 Model Evaluation: Diversity and Discrimination

This section re-examines the findings of previous feature distillation or mask feature reconstruction works illustrated in Sec. 1 and highlights their limitations in incorporating diversity and discrimination.

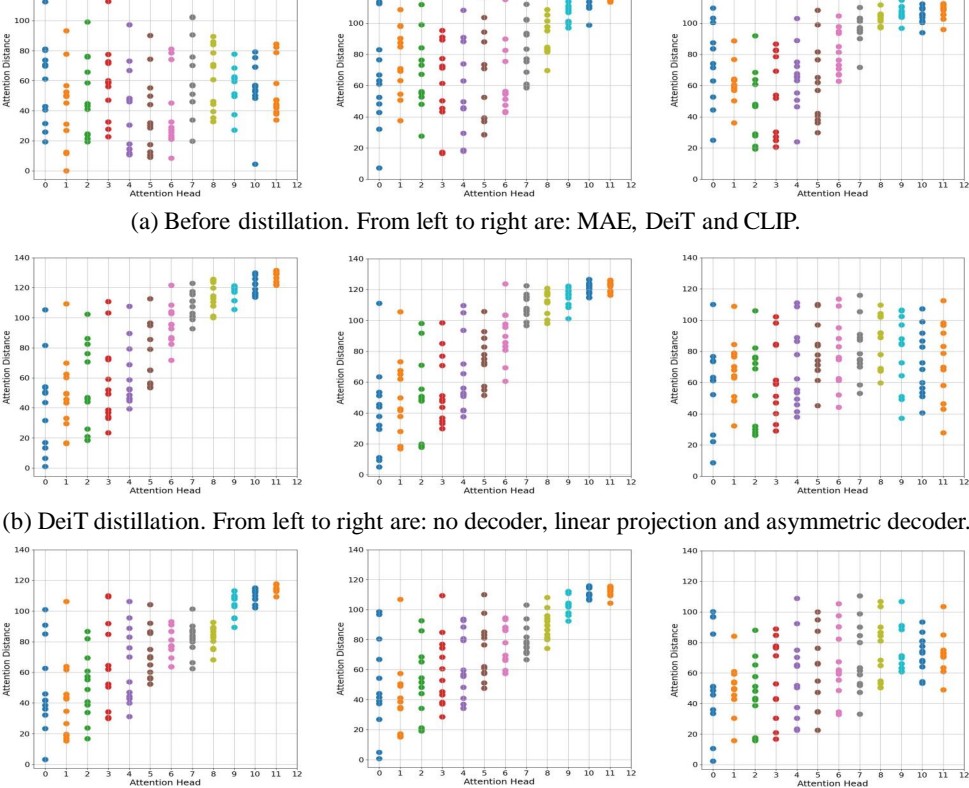

(a) Before distillation. From left to right are: MAE, DeiT and CLIP.

(b) DeiT distillation. From left to right are: no decoder, linear projection and asymmetric decoder.

(c) CLIP distillation. From left to right are: no decoder, linear projection and asymmetric decoder.

Figure 1: Average head distance after feature distillation with various decoders. (a) are the baselines. (b) use the supervised DeiT model as the teacher. (c) use the CL-based CLIP model as the teacher.

## 2.1 Preliminary

We first introduce the definitions of diversity and discrimination and the evaluation strategies we used. **Discrimination** means that the representations contain more global patterns tailored to object shapes, which is beneficial for recognizing objects and distinguishing images. **Diversity** is a relative concept, which means that the model pays more attention to local information and can achieve more evenly distributed representations, particularly in the last few layers.

We measure these properties by **average head distance** [41, 10] and **normalized mutual information (NMI)** [33]. The former calculates the average distance between the query tokens and the key tokens based on their attention weights, providing insight into whether the attention is global or local. The latter measures whether the attention is attending to different tokens or similar ones and is calculated following [27]. Specifically, let a uniform distribution $p(q) = \frac{1}{N}$ represent the distribution of query tokens, where $N$ is the total token number. The joint distribution of query and key is then computed as $p(q, k) = \pi(k|q)p(q)$, where $\pi(k|q)$ is the normalized self-attention matrix. Thus, NMI can be calculated by $\frac{I(q,k)}{\sqrt{H(q)H(k)}}$ where $I(\cdot, \cdot)$ is the mutual information and $H(\cdot)$ is the marginal entropy.

## 2.2 The Increase in Diversity Derives from the Asymmetric Designs

Fig. 1 measures the average head distance after feature distillation with a consistent encoder structure (vanilla Vision Transformer (ViT) [10]) for both the teacher and student models, along with various decoders only for the student. It can be seen that when the encoder is kept the same, using no decoder or linear projection decoder leads to a negligible increase (or even decrease) in attention diversity, reflecting that feature distilling itself cannot bring benefits to diversity. Adding some extra attention layers to the decoder can make the student encoder more diverse, but it hinders discrimination since the last layer no longer captures long-range patterns. Fig. 2(a) further compares NMI using the DeiT teacher and the results are in line with the attention visualization, *i.e.*, without asymmetric designs,

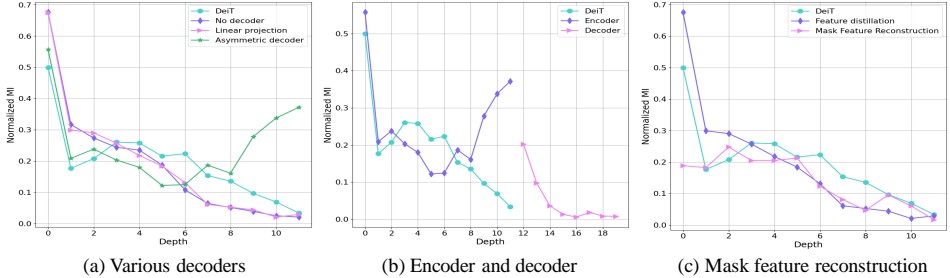

Figure 2: The normalized mutual information (NMI) of (a) various decoders, (b) encoder and decoder, and (c) mask feature reconstruction.

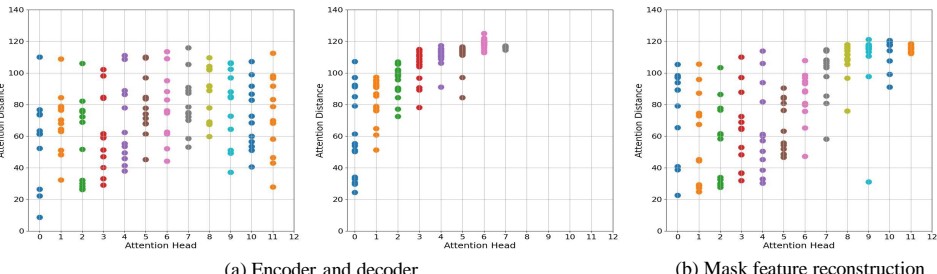

Figure 3: Average head distance of (a) encoder and decoder, and (b) mask feature reconstruction.

the student collapses into homogeneity and pays attention to similar tokens in the last few layers. Conversely, the use of asymmetric decoders greatly reduces discrimination.

The above discussions focus on varying decoders, while FD [41] introduces asymmetric designs to the encoder by adding additional learnable parameters and relative position bias to the attention layers of the student. In the appendix, we demonstrate that the increase in diversity observed in FD also arises from these designs and the diversity brought by them is not always significant.

## 2.3 The Asymmetric Decoder Harms the Encoder Discrimination

Fig. 3(a) and Fig. 2(b) further measure the average head distance and NMI of the asymmetric decoder. Our findings suggest that the decoder has transferred the discrimination of the teacher, as its behavior is similar to that of the last few layers of the teacher model where attention collapse occurs. Reducing the number of decoder layers does not eliminate this transfer, as further demonstrated in the appendix. Since only the student encoder is retained and applied to downstream tasks after distillation, the semantic information that the model maintained is weakened, which explains why in dBOT, different teachers tend to yield similarly-behaving models after multi-stage distilling. Note that dBOT conducts feature distilling in a mask reconstruction way, while we demonstrate in both Sec. 2.4 and the visualization in the appendix that it behaves similarly to directly distilling features.

## 2.4 Mask Reconstruction of High-Level Semantics Does not Help Improve Diversity

Fig. 3(b) and Fig. 2(c) examine the influence of mask reconstructing high-level information. To eliminate the effect of the asymmetric decoder, we feed both the masks and tokens into the encoder simultaneously and use only linear projection as the decoder. The overall process is thus similar to SimMIM [44], except that we use the high-level information obtained from the supervised/CL teacher as the distilling objective. Fig. 3(b) proves that reconstructing high-level information brings no diversity gains towards directly distilling features, which is consistent with the finding of [45], *i.e.*, reconstruction is unnecessary for MIM with semantic-rich teachers. This phenomenon also implies that the diversity of MIM mainly arises from the low-level reconstructing objective rather than from the reconstruction itself, since diversity is absent in high-level reconstruction.

## 3 Hybrid Distillation

From the above discussion, we conclude that existing distillation pipelines have limitations in providing discrimination and diversity. Thus, we further propose a novel hybrid distillation framework to ensure these important properties, and this section elaborates on its details.

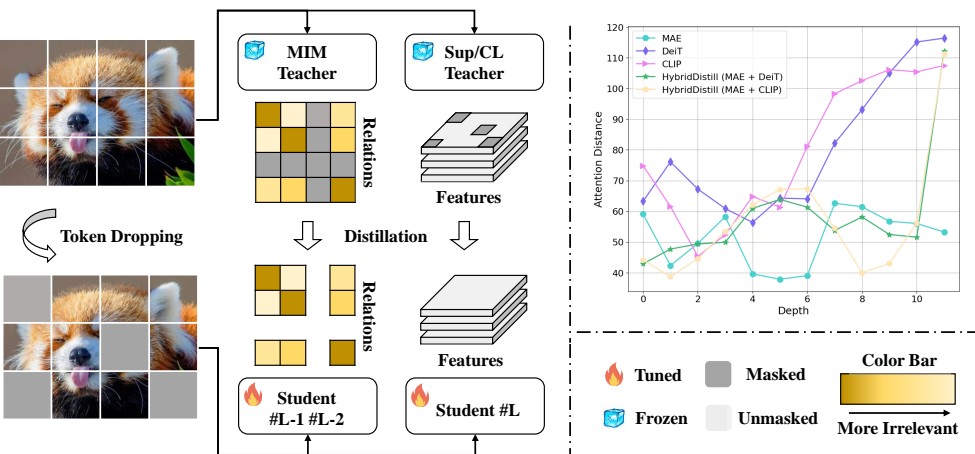

Figure 4: Hybrid Distill pipeline and its effectiveness in ensuring discrimination and diversity.

## 3.1 Overview

Given a supervised/CL pre-trained model $T_c$, and a MIM pre-trained model $T_m$, Hybrid Distill simultaneously distills knowledge from these two different types of pre-trained teachers, aims at combining their respective advantages to enhance the new representations in a randomly initialized student model $S_\theta$ where $\theta$ is its learnable parameters. ViT [10] is adopted for all the models in Hybrid Distill, and $T_m$ is provided by MAE [13] while $T_c$ is provided by DeiT [36] or CLIP [30].

Specifically, the Hybrid Distill framework is shown in Fig. 4 and its overall objective is:

$$
\begin{aligned}
\max_\theta \ \mathbb{E}_{x\sim\mathcal{X}} \ & \mathcal{D}\left\{T_c(x)\odot M, S_\theta(M\odot x)\right\} \\
& + \alpha\mathcal{D}\left\{T'_m(x)\odot M, S'_\theta(M\odot x)\right\},
\end{aligned}
\tag{1}
$$

where $\odot$ is an element-wise product operation. $M$ is a mask provided by the teacher model using the strategy described in Sec. 3.2 and $M\odot x$ denotes the unmasked patches. $\mathcal{D}(\cdot,\cdot)$ is the distance measurement, and we use smooth L1 distance in our experiment. $\alpha$ is the hyperparameter that controls the contribution of the two teacher models. Note that we do not distill the final output features $T_m(x)$ for the MIM pre-trained model but instead use the token relations in the previous ViT layers, denote as $T'_m(x)$, as the learning objective. Details are illustrated in Sec. 3.2.

## 3.2 Distilling Strategies

**What to distill?** Different from previous works [41, 11, 45] that directly distill the features of teacher models, we analyze that the diversity of MIM pre-trained models arises from their superior token-level relationship modeling, while supervised/CL pre-trained models excel at image-level discrimination. Hence, we apply different distilling targets to $T_c$ and $T_m$ to fully utilize their respective advantages. Specifically, taking $T_m$ as an example, we decompose $T_m$ into $T_m^1 \circ T_m^2 \circ \cdots \circ T_m^L$, where $T_m^i$ is the $i^{th}$ layer of $T_m$ and is composed of a multi-head self-attention (MSA) layer and an MLP layer. Given $x_m^i$ as the input of the $i^{th}$ layer, the calculation in $T_m^i$ can be represented as:

$$
\begin{aligned}
\mathrm{R}_m^i(x_m^i) &= Q_m^i(x_m^i)K_m^i(x_m^i)^T, \\
\mathrm{MSA}_m^i(x_m^i) &= \mathrm{Softmax}\left(\mathrm{R}_m^i(x_m^i)/\sqrt{d}\right)V_m^i(x_m^i), \\
T_m^i(x_m^i) &= x_m^i + \mathrm{MLP}(x_m^i + \mathrm{MSA}_m^i(x_m^i)),
\end{aligned}
\tag{2}
$$

where $Q_m^i$, $K_m^i$, and $V_m^i$ denotes the linear mappings for $x_m^i$ and $d$ equals to the dimension of $x_m^i$. Then, for MIM pre-trained model $T_m$, we set the token relation $\mathrm{R}_m^i(x_m^i)$ as the distilling target, while for supervised/CL pretrained model $T_c$, we set the output features $T_c^i(x_c^i)$ as the target.

**Where to distill?** As shown in Fig. 1(a), supervised and CL models tend to collapse into homogeneity in the last few layers, so Hybrid Distill chooses to distill token relations from $T_m$ in these layers to address this collapse and improve diversity. While for the last layer of $S$ which is

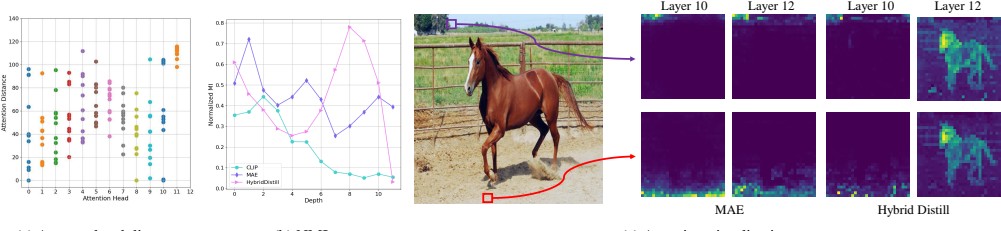

(a) Average head distance      (b) NMI      (c) Attention visualization

Figure 5: The (a) average head distance, (b) NMI, and (c) attention visualization of the student model obtained from Hybrid Distill with MAE and CLIP teachers.

crucial for discrimination, Hybrid Distill directly distills knowledge from $T_c$ using the output features. Specifically, we distill token relations from $T_m$ at the $L-1$ and $L-2$ layers and distill features from $T_c$ at the $L$ layer of ViT. Accordingly, the learning objective $T_c(x)$ and $T'_m(x)$ in Eq. 1 become:

$$
\begin{aligned}
T_c(x) &= T_c^L(x), \\
T'_m(x) &= [R_m^{L-1}(x), R_m^{L-2}(x)].
\end{aligned}
\tag{3}
$$

**Distillation acceleration via redundant token dropping.** Suppose the input is divided into $N$ tokens, *i.e.*, $x \in \mathbb{R}^{N \times d}$, Hybrid Distill can directly distill token relations and features using all the $N$ tokens. However, since some tokens in the image may be redundant, it is promising to mask these tokens for the student model $S$ to reduce memory and time costs. Furthermore, removing redundant tokens can play a regulatory role, helping the model avoid local optima during the distillation process.

Specifically, we use the MIM pre-trained teacher $T_m$ to guide the identification of redundant tokens and provide the token mask. Inspired by [20], we propose a progressive redundant token masking strategy, which generates token masks at different layers of $T_m$ in a progressive manner. Given $x_m^i$ and the mask $M_m^{i-1}$ provided by the previous layer, we define the tokens in $x_m^i \odot M_m^{i-1}$ and are top $K\%$ similar to their average token as redundant tokens in the $i^{th}$ layer and generate a redundant token mask for them. The above process is denoted as $T(x_m^i \odot M_m^{i-1}, K)$. Next, we update $M_m^i$ using $T(x_m^i \odot M_m^{i-1}, K)$ and $M_m^{i-1}$ as follows:

$$
M_m^i = \begin{cases} M_m^{i-1} - T(x_m^i \odot M_m^{i-1}, K), & \text{if } i \in I, \\ M_m^{i-1} & \text{if } i \notin I. \end{cases}
\tag{4}
$$

where $I$ is the set of layers required to update the token mask. For $M_m^0$, all elements are set to 1. Finally, we set the mask $M$ for the student model as $M = M_m^L$.

### 3.3 Property Analysis

**Average head distance.** Fig. 5(a) visualizes the average head distance of the student model with CLIP and MAE as teachers, while the visualization of CLIP and MAE teachers themselves are included in Fig. 1(a). These visualizations demonstrate that Hybrid Distill enhances the discrimination ability of the student model, compensating for the semantic lacking problem of the MAE teacher. Moreover, Hybrid Distill avoids succeeding attention collapse from the CLIP teacher and generates more diverse representations in the last few layers.

**Normalized mutual information.** Fig. 5(b) further inspects the NMI. The results demonstrate that the mutual information between tokens is significantly enhanced in the layers where the MAE token relationships are distilled. Besides, this enhancement does not compromise the discrimination obtained from CLIP, as evidenced by attention in the final layers still attending to similar tokens.

**Attention visualization.** Fig. 5(c) further visualizes the attention between a given query and other keys at different layers to examine behaviors. Compared to MAE, Hybrid Distill exhibits better discrimination ability, *i.e.*, the query tokens of the last layer have global attention towards the main object of the images, regardless of their location. Besides, Hybrid Distill also improves the locality of the model in the $10^{th}$ layer, where attention collapse is known to occur in the CLIP teacher.

### 3.4 Discussion with Other Distillation Methods

Compared to previous distillation methods [41, 11, 23, 40, 29], Hybrid Distill stands out by not being restricted to using a single teacher network. In addition to addressing the limitations of single-teacher

Table 1: Main results on ImageNet-1k classification, COCO detection and instance segmentation, and ADE20K semantic segmentation. $\star$: using MAE+DeiT teachers. $\dagger$: using MAE+CLIP teachers.

| Method | Backbone | Distill. | IN-1K | COCO | | ADE20K |
| --- | --- | --- | --- | --- | --- | --- |
| | | | | $AP^{box}$ | $AP^{Mask}$ | |
| DeiT [36] | | | 81.8 | 46.9 | 41.5 | 47.0 |
| MoCo v3 [7] | | | 83.2 | 45.5 | 40.5 | 47.1 |
| DINO [2] | | | 83.3 | 46.8 | 41.5 | 47.2 |
| MAE [13] | ViT-B | | 83.6 | 48.4 | 42.6 | 48.1 |
| CAE [5] | | | 83.3 | 48.0 | 42.3 | 47.7 |
| SdAE [8] | | | 84.1 | 48.9 | 43.0 | 48.6 |
| CLIP [30] | | | 83.6 | 47.6 | 42.3 | 49.6 |
| MAE [13] | ViT-L | | 85.9 | 54.0 | 47.1 | 53.6 |
| CLIP [30] | | | 86.1 | 52.7 | 46.2 | 54.2 |
| Distill-DeiT | | | 82.0 | 47.7 | 42.1 | 47.3 |
| Distill-MAE | ViT-B | ✓ | 83.7 | 49.1 | 43.1 | 47.8 |
| Distill-CLIP | | | 84.8 | 49.5 | 43.5 | 50.3 |
| Hybrid Distill$\star$ | ViT-B | ✓ | 83.7 | 50.3 | 44.2 | 49.1 |
| Hybrid Distill$\dagger$ | | | **85.1** | **50.6** | **44.4** | **51.5** |
| Hybrid Distill$\dagger$ | ViT-L | ✓ | **88.0** | **54.6** | **47.6** | **56.3** |

Table 2: Classification results on CIFAR100, Cars and INautralist19. $\star$: using MAE+DeiT teachers. $\dagger$: using MAE+CLIP teachers.

| Method | Backbone | CIFAR100 | Cars | INaturalist19 | Mean |
| --- | --- | --- | --- | --- | --- |
| DeiT [36] | ViT-B | 91.4 | 92.0 | 77.3 | 86.9 |
| MAE [13] | ViT-B | 89.6 | 89.5 | 75.2 | 84.8 |
| Distill-DeiT | ViT-B | 91.2 | 92.5 | 78.3 | 87.3 |
| Distill-MAE | ViT-B | 90.3 | 93.1 | 79.0 | 87.5 |
| Distill-CLIP | ViT-B | 91.6 | 94.3 | 81.6 | 89.2 |
| Hybrid Distill$\star$ | ViT-B | 91.7 | 94.1 | 80.2 | 88.7 |
| Hybrid Distill$\dagger$ | ViT-B | **92.0** | **94.5** | **81.9** | **89.5** |
| Hybrid Distill$\dagger$ | ViT-L | **94.5** | **95.6** | **85.3** | **91.8** |

distillation in enriching diversity (as discussed in Sec. 2), a more direct factor is that single-teacher distillation cannot create new knowledge, *e.g.*, creating additional discrimination for the student model when using the MIM teacher. Therefore, we believe that combining and utilizing existing knowledge from various teachers is more effective and convenient. Furthermore, with the growing availability of large-scale pre-trained models within the community, it becomes increasingly valuable to explore new ways to utilize these models and combine their strengths. This further enhances the practical value of our Hybrid Distill, and we hope our work would shed light on new directions.

# 4 Experiments

## 4.1 Implementation Details

Hybrid Distill is conducted on 8 V100 GPUs and is built on the codebase of dBOT [23], so most of its settings are in line with dBOT. Specifically, the batch size, learning rate, and weight decay are set to 1024 and 6e-4, and 0.05, respectively. AdamW [26] optimizer and cosine decay [25] schedule is used. The input size is $224^2$. For ViT-B, the distillation is based on ImageNet-1K and the epoch is 300 for main results and 100 for ablation studies. For ViT-L, the distillation is based on ImageNet-21K and the epoch is 40. The hyperparameter $\alpha$ is set to $1.0$ and the redundant token masking set $I$ is set to $[0, L/3, 2L/3]$ following [20]. The performances are tested on different downstream tasks. For classification, we report results on ImageNet-1K, CIFAR100 [19], Cars [18], and iNaturalist19 [37]. For object detection and instance segmentation, we fine-tune the student model on COCO [22] using Mask-RCNN [15] following [5]. For semantic segmentation, the evaluation is conducted on ADE20K [47] using the ViT with UperNet [42] following [5, 8]. More details are included in the appendix.

## 4.2 Main Results

This section presents benchmark results of Hybrid Distill on different downstream. We also list results for supervised and self-supervised pre-trained models, as well as 300-epoch uni-distillation baselines

Table 3: Different combinations of two teacher models. $T_c(x)$: DeiT, $T_m(x)$: MAE.

| Targets | $AP^{box}$ | $AP^{mask}$ |
|---|---|---|
| $T_c(x)$ | 47.5 | 41.8 |
| $T_m(x)$ | 48.9 | 43.1 |
| $T_c(x) + T_c'(x)$ | 46.8 | 41.5 |
| $T_m(x) + T_m'(x)$ | 48.9 | 43.2 |
| $T_c(x) + T_m'(x)$ | **50.0** | **43.9** |

Table 4: Different combinations of two teacher models. $T_c(x)$: CLIP, $T_m(x)$: MAE. $\star$: using the ImageNet-100 pretrained weights.

| Targets | $AP^{box}$ | $AP^{mask}$ |
|---|---|---|
| $T_c(x)$ | 49.1 | 43.1 |
| $T_m(x)$ | 48.9 | 43.1 |
| $T_c(x) + T_c'(x)$ | 49.1 | 43.2 |
| $T_c(x) + T_m'(x)$ | **50.4** | **44.1** |
| $T_c(x) + T_m'(x)^\star$ | 49.5 | 43.5 |

Table 5: The distilling targets of $T_m'(x)$. $T_c(x)$: DeiT, $T_m(x)$: MAE. $\star$ means distilling MAE and DeiT features at the last layer.

| Targets | $AP^{box}$ | $AP^{mask}$ |
|---|---|---|
| $T_m^{i\,\star}$ | 47.7 | 42.1 |
| $T_m^i$ | 49.6 | 43.5 |
| $MSA_m^i$ | 49.8 | 43.7 |
| $R_m^i$ | **50.0** | **43.9** |

Table 6: The distilling targets of $T_m'(x)$. $T_c(x)$: CLIP, $T_m(x)$: MAE.

| Targets | $AP^{box}$ | $AP^{mask}$ |
|---|---|---|
| $T_m^i$ | 49.9 | 44.0 |
| $MSA_m^i$ | 50.1 | 44.0 |
| $R_m^i$ | **50.4** | **44.1** |

which use the same symmetrical structures as Hybrid Distill, for comparison. As shown in Tab. 1, Hybrid Distill achieves performance gains on all downstream tasks, especially for the dense-level ones. Specifically, although the performance of DeiT is suboptimal, its strength can be complementary to MAE and brings considerable benefits, *i.e.*, when using DeiT and MAE teachers, Hybrid Distill achieves 50.3 $AP^{box}$ and 44.2 $AP^{mask}$ on COCO, as well as 49.1 mIoU on ADE20K, surpassing Distill-MAE by 1.2, 1.1, and 1.3, respectively. Similarly, Hybrid Distill achieves 50.6 $AP^{box}$ and 44.4 $AP^{mask}$ on COCO, as well as 51.5 mIoU on ADE20K when using CLIP and MAE teachers, outperforming Distill-CLIP by 1.1, 0.9, and 1.2, respectively. When using the VIT-L backbone, the performance can be further boosted to 54.6 $AP^{box}$, 47.6 $AP^{mask}$ and 56.3 mIoU on respective tasks. The improvement on ImageNet-1k is not significant, probably because the distillation is performed on the same dataset, thus increasing diversity fails to bring further gains. In Tab. 2, we further evaluate Hybrid Distill on several small-scale classification datasets and observe more significant gains.

## 4.3 Ablation Study

This section ablates different variants of Hybrid Distill. The results are reported on dense-level COCO detection and segmentation tasks, as diversity has a stronger influence on these dense-level tasks [27].

**Different combinations of two teachers.** We first evaluate the benefits of combining two teachers for distillation. As shown in Tab. 3, adding additional MAE attention regularization can bring noticeable improvements (2.5 on $AP^{box}$ and 2.1 on $AP^{mask}$) compared to directly distilling from the DeiT teacher. Moreover, the additional attention regularization cannot bring benefits when only using a single DeiT teacher, which suggests that the benefits come from the introduction of MAE teacher. The above conclusions are consistent when using CLIP and MAE teachers, as illustrated in Tab. 4. We also try a much weaker version of MAE teacher which is only pre-trained on ImageNet-100 for 100 epochs in Tab. 4. We lower the weight of this teacher to avoid its impact on discrimination. The results are still positive, which reflects the power of the MIM pre-training in modeling diversity.

**Distilling target of the MIM teacher.** We then examine the distilling target of the MIM teacher. As shown in Tab. 5, distilling the relation $R_m^i$ brings the best detection performance (50.0$AP^{box}$). Distilling $MSA_m^i$ achieves a close performance (49.8$AP^{box}$) since its essential is also distilling relationships, while directly distilling the feature maps $T_m^i$ brings the worst performance (49.6$AP^{box}$). Nevertheless, all these schemes outperform the DeiT distillation baseline, and the trends are consistent when using CLIP and MAE teachers, as shown in Tab. 6. Besides, we also evaluate a basic setting that directly distills the features of both the MAE and DeiT teachers at the last layer. The result is far from satisfactory, which highlights the effectiveness of the designs in Hybrid Distill.

**Distilling position of the MIM teacher.** Tab. 7 inspect the distilling position of the MIM teacher. We first experiment with distilling MAE relations at the front, middle, and back layers. Distilling at the back layers achieves better results, *i.e.*, 1.5$AP^{box}$ and 2.4$AP^{box}$ gains towards distilling at the

Table 7: The distilling position of $T_m$.

| Distilling layers | $AP^{box}$ | $AP^{mask}$ |
|---|---|---|
| 1-11 | 48.8 | 43.0 |
| 1,2,3 | 47.4 | 41.9 |
| 5,6,7 | 48.3 | 42.7 |
| 9,10,11 | 49.8 | 43.7 |
| 10,11 | **50.0** | **43.9** |
| 11 | 49.2 | 43.3 |

Table 8: The token masking strategy.

| Strategy | Ratio | $AP^{box}$ | $AP^{mask}$ |
|---|---|---|---|
| No | $100\%$ | **50.0** | **43.9** |
| Random | $35\%$ | 49.2 | 43.3 |
| Direct | $35\%$ | 49.6 | 43.7 |
| Progressive | $13\%(50\%^3)$ | 48.4 | 42.8 |
| Progressive | $34\%(70\%^3)$ | 49.9 | 43.8 |
| Progressive | $73\%(90\%^3)$ | 49.9 | 43.8 |

front and middle, respectively. The results are consistent with the fact that attention collapse tends to occur in these back layers. We then ablate the number of distilling layers and find that distilling at the two layers preceding the final layer (*i.e.*, 10,11) contributes to the best results.

**Token masking strategy.** Tab. 8 studies different masking strategies for the student model. Since we progressive drop the redundant tokens three times, the actual tokens used in the student model are $(1 - K)^3\%$. We observe that when dropping $30\%$ tokens at a time, Hybrid Distill achieves very close performance ($49.9AP^{box}$ and $43.8AP^{mask}$) to the no masking results and outperforms the random masking strategy and the direct masking strategy which only generates token mask at the last layer. In addition, we notice that our token masking strategy also has a regularizing effect, which can prevent the model from falling into a locally optimal when training for longer epochs. Details about this effect are included in the appendix.

## 5 Related Work

**Representation learning.** Pre-training on large-scale datasets (e.g., ImageNet [32], JFT [34], Kinetics [3], etc.) is typically utilized for downstream initialization. Except for the common supervised pre-training [16, 10, 24], contrastive learning (CL) [4, 14, 6, 12] and masked image modeling (MIM) [1, 44, 13] dominate the recent research. The former is achieved by pulling close the features of two different augment views of the input image. While the latter, inspired by masked language modeling [17, 46] in NLP, is realized by reconstructing the mask part of the input image. Recently multi-model extensions [30, 9, 21] of the CL pre-training have also been proposed by utilizing the paired text description of the given image. These different types of pre-training frameworks are proven to have different properties [27, 43], and this paper aims to combine their respective excellent properties to boost a student model.

**Knowledge distillation.** Knowledge distillation [28, 35, 31] utilizes a well-trained teacher to guide the feature learning of the student model, thus transferring its ability to the student. Beyond its success in supervised learning, some recent works [41, 11, 39, 40, 29] utilize it to extend existing pretrained models or paradigms. Feature distillation (FD) [41] finds that distilling the feature map of the supervised/CL pretrained teacher can bring diverse representation to the student and make it more friendly for downstream fine-tuning. dBOT [23], MVP [40], and BEiT v2 [29] change the mask reconstruction object of MIM to the knowledge of the teacher model to boost MIM pre-training with semantic information. In this paper, we analyze their properties and propose a new hybrid distillation framework to deal with their deficiencies.

## 6 Conclusion

This paper proposed a hybrid distillation framework that simultaneously distills knowledge from both the supervised/CL pre-trained teacher and MIM pre-trained teacher to enhance the diversity and discrimination of the student. The framework addresses the limitations of single-teacher distillation, where increasing diversity through the use of asymmetric designs may harm discrimination. Specifically, Hybrid Distill carefully designs the distilling target and location, *i.e.*, distilling relations from MIM in layers where attention collapse tends to occur and distilling features from supervised/CL in the last layer to preserve discrimination. A progressive redundant token masking strategy is also proposed for reducing the distilling costs. Experiments prove that Hybrid Distill can acquire better properties and achieve promising results on various downstream. We hope our research would shed light on a new direction for applying existing large-scale pre-trained models.

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
