# Appendix for Hybrid Distillation: Connecting Masked Autoencoders with Contrastive Learners

Table A1: Compared with more baselines using ViT-B as the backbone. $\star$: using MAE+DeiT teachers. $\dagger$: using MAE+CLIP teachers.

| Method | COCO | | ADE20K |
|---|---|---|---|
| | $AP^{box}$ | $AP^{Mask}$ | |
| Distill-DeiT | 47.7 | 42.1 | 47.3 |
| Distill-MAE | 49.1 | 43.1 | 47.8 |
| Distill-CLIP | 49.5 | 43.5 | 50.3 |
| FD-DeiT [11] | 47.0 | 41.6 | 47.9 |
| FD-MAE [11] | 48.1 | 42.6 | 47.0 |
| FD-CLIP [11] | 49.2 | 43.3 | 50.5 |
| dBOT-DeiT [9] | 47.5 | 41.9 | 47.9 |
| dBOT-MAE [9] | 49.3 | 43.5 | 48.2 |
| Hybrid Distill$^{\star}$ | 50.3 | 44.2 | 49.1 |
| Hybrid Distill$^{\dagger}$ | **50.6** | **44.4** | **51.5** |

## A    More Experimental Results

### A.1    Compared with More Baselines

Tab. A1 compares Hybrid Distill with two other methods, *i.e.,* dBOT [9] and FD [11], which employ asymmetric designs in distillation. We conduct distilling for 300 epochs based on their corresponding official codes[1]. We omit the dBOT-CLIP result since dBOT specifically removes the asymmetric designs for CLIP, thus its distillation process is similar to our Distill-CLIP baseline. As shown in Tab. A1, their benefits towards symmetrical distillation are not always significant, and the performance is inferior to our Hybrid Distill, which validates the effectiveness of our framework.

### A.2    Results with Cascade Mask-RCNN

Tab. A2 further presents the object detection and instance segmentation results of Hybrid Distill with Cascade Mask-RCNN, which allows for a direct comparison with dBOT [9], as they also provide 1600-epoch distillation results under this setting. As shown, 300-epoch Hybrid Distill with MAE and DeiT teachers can achieve 53.0 $AP^{box}$, outperforming 1600-epoch dBOT-DeiT (52.5 $AP^{box}$) and dBOT-MAE (52.7 $AP^{box}$). Additionally, 300-epoch Hybrid Distill with MAE and CLIP teachers achieves 53.4 $AP^{box}$, which is also very close to the 1600-epoch dBOT-CLIP result (53.6 $AP^{box}$). The above results reflect that due to the better properties obtained, Hybrid Distill can obtain promising results with fewer training epochs.

---

[1]dBOT [9]: https://github.com/liuxingbin/dbot/. FD [11]: https://github.com/SwinTransformer/Feature-Distillation/. Since FD does not provide codes for downstream verification, we uniformly perform verification under our downstream frameworks.

Submitted to 37th Conference on Neural Information Processing Systems (NeurIPS 2023). Do not distribute.

Table A2: Object detection and instance segmentation results with *Cascade Mask-RCNN*. $\star$: using MAE+DeiT teachers. $\dagger$: using MAE+CLIP teachers.

| Method | Epoch | $AP^{box}$ | $AP^{mask}$ |
|---|---|---|---|
| Distill-DeiT | 300 | 50.4 | 43.4 |
| Distill-MAE | 300 | 51.9 | 44.7 |
| Distill-CLIP | 300 | 52.4 | 45.0 |
| dBOT-DeiT [9] | $2 \times 800$ | 52.5 | - |
| dBOT-MAE [9] | $2 \times 800$ | 52.7 | - |
| dBOT-CLIP [9] | $1 \times 1600$ | 53.6 | - |
| Hybrid Distill$^{\star}$ | 300 | 53.0 | 45.6 |
| Hybrid Distill$^{\dagger}$ | 300 | **53.4** | **45.9** |

Table A3: Hybrid Distill uses MAE and DINO as teachers. Object Detection and instance segmentation results are reported with *Mask-RCNN*, following the setting in Tab. 1 of our main paper.

| Method | $AP^{box}$ | $AP^{mask}$ |
|---|---|---|
| MAE | 48.4 | 42.6 |
| DINO | 46.8 | 41.5 |
| Distill-DINO | 47.5 | 41.9 |
| Distill-MAE | 49.1 | 43.1 |
| Hybrid Distill | **49.6** | **43.5** |

## A.3 Hybrid Distillation with DINO

Tab. A3 test the results of our Hybrid Distill using the MAE and DINO teachers. Under this setting, Hybrid Distill achieves 49.6 $AP^{box}$ and 43.5 $AP^{mask}$. Although still superior to the baselines, results with DINO are not as good as those with CLIP and DeiT. We analyze that this is because the discrimination of DINO is weaker than DeiT and CLIP, which makes its complementarity with MAE also weaker than the latter two. The visualization in Fig.A1 provides evidence for this. On the one hand, we notice that the average attention distance of DINO itself is lower than that of DeiT and CLIP in the final layer. On the other, the attention maintenance of the final layer after distillation is weaker compared with that obtained by DeiT and CLIP.

## A.4 More Ablation Studies

**The choice of hyperparmeter $\alpha$.** Tab. A4 ablates different setting of $\alpha$. It can be concluded that adding additional MIM supervision can lead to performance improvement towards not using MIM supervision ($\alpha = 0$), regardless of the value of $\alpha$. While setting $\alpha$ to 1.0 can bring the best performance for both MAE+DeiT and MAE+CLIP teachers. Using the CLIP teacher achieves more stable performance since CLIP itself has higher quality compared with DeiT, while DeiT relies more on the help of MAE.

**Token masking strategy and local optima.** Tab. A5 further reveals that the proposed progressive redundant token masking strategy in Hybrid Distill can prevent the student from falling into local optima. As shown, when the token mask is removed and the distillation epoch is prolonged from 100 to 300, no performance gains are observed. This phenomenon has also been observed in [4]. We analyze that over-fitting is the root cause of this problem and introducing token masks can alleviate it since they can play a regulatory role. The performance gains achieved by the token masks provide clear support for their effectiveness.

# B Further Discussion about Diversity and Discrimination

## B.1 Asymmetric Encoder Designs

Fig. A2 studies the asymmetric encoder designs used in FD, *i.e.*, adding additional learnable parameters and relative position bias to the attention layers of the student. As shown, the asymmetric encoder (Fig. A2(c)) *de facto* improves diversity compared to using only the symmetric encoder (Fig. A2(b)).

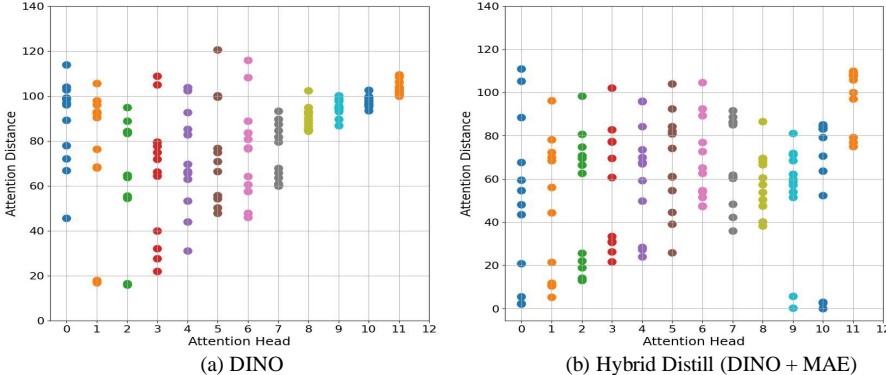

|  | (a) DINO | (b) Hybrid Distill (DINO + MAE) |

Figure A1: Average head distance of different (a) DINO baseline and (b) Hybrid Distill with MAE and DINO as teachers.

Table A4: Ablation on the hyperparameter $\alpha$ which controls the contribution of two teacher models.

(a) $T_c(x)$: DeiT, $T_m(x)$: MAE.

| $\alpha$ | 0 | 0.1 | 0.3 | 0.5 | 0.7 | 1.0 |
|---|---|---|---|---|---|---|
| $AP^{box}$ | 47.5 | 48.2 | 49.3 | 49.3 | 49.5 | **50.0** |
| $AP^{mask}$ | 41.8 | 42.6 | 43.4 | 43.4 | 43.5 | **43.9** |

(b) $T_c(x)$: CLIP, $T_m(x)$: MAE.

| $\alpha$ | 0 | 0.1 | 0.3 | 0.5 | 0.7 | 1.0 |
|---|---|---|---|---|---|---|
| $AP^{box}$ | 49.1 | 49.9 | 49.8 | 50.1 | 50.2 | **50.4** |
| $AP^{mask}$ | 43.1 | 43.8 | 43.8 | 43.9 | 44.1 | **44.1** |

Table A5: The token masking strategy for alleviating over-fitting. $\star$: using MAE+DeiT teachers. $\dagger$: using MAE+CLIP teachers.

| Method | Epoch | Masking | $AP^{box}$ | $AP^{mask}$ |
|---|---|---|---|---|
| Hybrid Distill$^{\star}$ | 100/300 |  | **50.0**/50.0 | **43.9**/44.0 |
| Hybrid Distill$^{\star}$ | 100/300 | ✓ | 49.9/**50.3** | 43.8/**44.2** |
| Hybrid Distill$^{\dagger}$ | 100/300 |  | **50.4**/50.2 | **44.1**/44.1 |
| Hybrid Distill$^{\dagger}$ | 100/300 | ✓ | 50.2/**50.6** | 43.9/**44.4** |

However, compared to the DeiT teacher (Fig. A2(a)), it does not bring noticeable diversity gains. Therefore, we conclude that the diversity brought by the asymmetric encoder is not always significant.

## B.2    Mask Feature Reconstruction in dBOT

Fig. A3 compares two variants of dBOT, *i.e.*, with the same asymmetric decoder design but conducting direct feature distillation and mask feature reconstruction, respectively. It can be seen that the two tasks bring no significant differences, *i.e.*, the diversity is increased and the discrimination is lost regardless of the task. These visualizations further support our claim in Sec. 2.3 and Sec 2.4 of our main paper.

## B.3    Reducing the Number of the Asymmetric Decoder Layers

Fig. A4 investigates the effect of reducing the number of asymmetric decoder layers. We find that even with a reduced number of decoder layers, the discrimination in the last layer of the encoder still cannot be maintained. Therefore, we abandon this asymmetric decoder design in our Hybrid Distill to avoid losing discrimination.

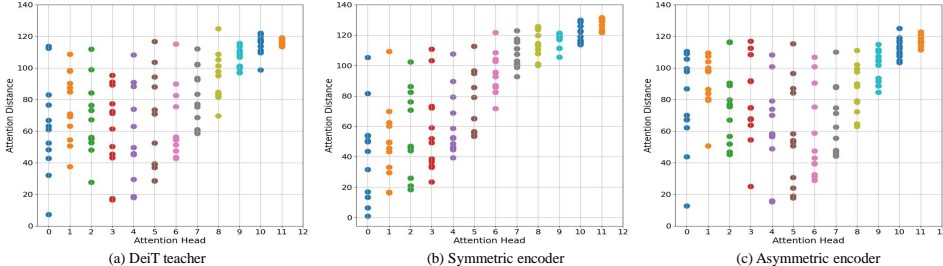

Figure A2: Average head distance of (a) DeiT teacher and student models with (b) symmetric encoder and (c) asymmetric encoder.

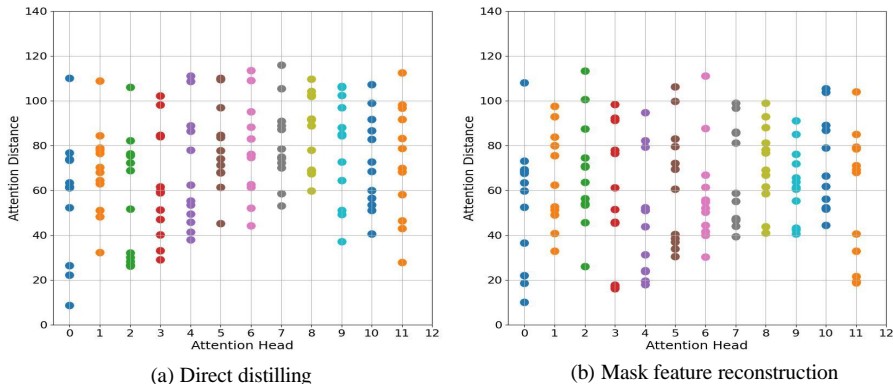

Figure A3: Average head distance of different dBOT variants that conduct (a) direct feature distillation and (b) mask feature reconstruction, respectively.

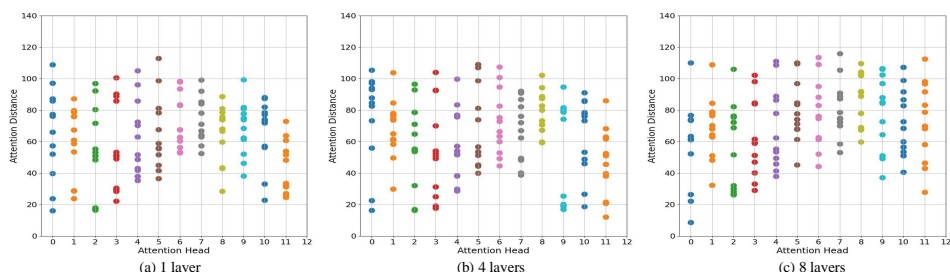

Figure A4: Average head distance of using (a) 1, (b) 4, and (c) 8 asymmetric decoder layers, respectively.

## C  Implementation Details for Different Downstream Tasks

**Classification.**    We report the fine-tuning results on ImageNet-1K. Following dBOT [9], the learning rate is set to 3e-4 and the batch size is set to 256. We also report results on CIFAR100 [7], Cars [6], and iNaturalist19 [10]. For these datasets, the batch size is 768 and the learning rate is 7.5e-6.

**Object detection and instance segmentation.**    Following [1], we fine-tune the student model on COCO [8] using the Mask-RCNN [5] framework. We train the network with the 1x schedule and the learning rate is set to 3e-4 for ViT-B and 2e-4 for ViT-L. We also provide the 1x results using the Cascade Mask-RCNN framework in the appendix, and the learning rate is set to 3e-4.

**Semantic segmentation.**    The semantic segmentation evaluation is conducted on ADE20K [13]. Following [1, 2], we use ViT [3] with UperNet [12] framework and fine-tune the model for 160k iterations. The batch size, learning rate, and weight decay are set to 16, 4e-4, and 0.05, respectively.

## D   Limitation

Hybrid Distill jointly utilizes two teacher models to guide the representation learning of the student. Although exhibiting promising properties and results, the additional overhead of introducing two teachers may be a limitation. Fortunately, since the teacher model does not require gradient updates, the training cost of Hybrid Distill does not increase significantly, *i.e.*, the training time of Hybrid Distill with ViT-B backbone is around 1.2 times longer than that of using a single teacher. Besides, Hybrid Distill can achieve better performance with much fewer training epochs, as shown in Tab. A2. From this perspective, Hybrid Distill in turn reduces the training cost. Another possible limitation is that Hybrid Distill does not improve CLIP as much as DeiT after introducing the MAE teacher, and we analyze that it may be caused by the gap between the pre-training capacities of CLIP and MAE teachers. We look forward to better MIM models that can further facilitate our work.

## E   Reproducibility

We will release our source code once this paper is accepted.