# OpenReview forum: "Hybrid Distillation: Connecting Masked Autoencoders with Contrastive Learners"
_NeurIPS.cc/2023/Conference — Submitted to NeurIPS 2023_

### Official Review · Reviewer_yZYM · 2023-06-22

**Soundness:** 3 good
**Presentation:** 2 fair
**Contribution:** 3 good
**Rating:** 4
**Confidence:** 4

**Summary:**

This paper introduces a new hybrid distillation method for Vision Transformer. It is hybrid in the sense that two teacher models pre-trained by two different approaches, namely Contrastive Learning and Masked Image Modeling, are adopted in the distillation. In this way, the student model inherit both the diversity from MIM pre-trained teacher model and the discriminability from the CL pre-trained teacher model. Further more, a progressive masking strategy is adopted to reduce redundancy during distillation. Experiments on multiple popular classification/detection/segmentation benchmarks demonstrate the effectiveness of the proposed approach.

**Strengths:**

The idea of adopting two homgeneous teacher models is interetsing. Intuitively, a good design allows the student model to learn the advantages of both teacher models.

**Weaknesses:**

1. I find many experimental details in the explorative model evaluation part (section 2) are missing, making it hard for readers to assess the soundness of those experiments. Note that these experiments are crucial for justifying the motivation of this work. My questions are listed as follows:
a) What kind of distillation approach is used here? Is it the same as the one used in Hybrid Distillation, or just naive feature distillation? Also, details like where to distill and distill objectives are missing.
b) Why are the average head distance and normalizedmutual information good indicators of the discrimination power and diversity of the learnt representations? And how are the so called `discrimination and diversity` related to the downstream performances? So far I have only seen very vague definition of these two metrics, and I am not sure if they are solid proofs that teacher models pre-trained by different objectives do have different advantages that the student must learn.
c) The notations need further clarification. For instance, in Figure 2(a), the caption says 'various decoder' , indicating the curves are the values of NMI of the decoder. While there is actually a curve for `no decoder`, indicating the curves are the values of NMI of the encoder. These notations are very confusing.
d) The authors claim that `The Increase in Diversity Derives from the Asymmetric Designs`, but there are no ablation on if the symmetric design of MIM architecture would lead to the same diversity. The closet design is in section 2.4, but its objective is to reconstruct high-level features from the teacher model.

2. Similarly, some implementation details about the Hybrid Distillation itself are missing. For example, I do not believe the student is solely trained by the distillation objective. Is there also a supervised training objective like cross-entropy loss? The authors should make it more clear.

3. I wonder how did the authors get the results of baseline methods. For example, in table 1, CLIP ViT-B achieves an accuracy of 83.6 on IN-1K, and CLIP ViT-L achieves an accuracy of 86.1. These numbers are too good to be zero-shot results, so I have to assume they are fine-tuning results. Yet, based on [1], fine-tuning CLIP ViT-B achieves an accuracy of 85.7, and CLIP ViT-L achieves an accuracy of 88.0 on IN-1K, which have achieved better or on par performance comapred to the Hybrid Distillation (85.1 for ViT-B, 88.0 for ViT-L). In this case, the proposed Hybrid Distillation approach does not seems to show enough empirical gain.

4. As already discussed in the limitation section in the supplemental material, the gain of Hybrid Distillation over Distill-CLIP is not so significant (only 0.3\% on IN-1K with ViT-B).

References
[1] Dong X, Bao J, Zhang T, et al. CLIP Itself is a Strong Fine-tuner: Achieving 85.7% and 88.0% Top-1 Accuracy with ViT-B and ViT-L on ImageNet[J]. arXiv preprint arXiv:2212.06138, 2022.

**Questions:**

See weakness.

**Limitations:**

The authors have included discussion on the limited gain over Distill-CLIP.

---

> ### Author Rebuttal · Authors · 2023-08-10
>
> **Q1: [What kind of distillation approach is used in Section 2.]:**
>
> **Our distillation setup remains consistent with [1][2]**, where only the features in the last layer of the ViT teacher are utilized as distillation objectives. We introduce various decoders, including linear projection and transformer decoders, in addition to the scenario without a decoder, to examine their impact on the feature representation. Subsequently, we compute the L1 loss between the decoded features of the student model and the features extracted from the teacher model.
>
> **Q2: [The average head distance and normalized mutual information.]:**
>
> i) The per-head average attention distance, which is also used in [1][2][3][4], serves as an indicator of the attention range. When both long-range and short-range attention occurs within the same layer, tokens have the opportunity to gather diverse knowledge from other tokens, leading to increased diversity. **This notion of diversity is similarly defined and discussed in [2][3][4]**.
>
> ii)  The NMI metric, utilized in [4] as well, assesses the relationship between query tokens and key tokens. A low NMI value indicates that attention maps are less reliant on the query tokens, implying that all queries focus on similar key tokens. These key tokens often correspond to the main subjects in the images (e.g., as illustrated in Figure 5 of our paper, tokens from different positions all pay attention to the image's subject (horse)). We believe that this subject-focused ability is advantageous for discriminative tasks and define this ability as discrimination. **A similar definition of discrimination can also be found in [14]**.
>
> iii) As illustrated in Lines 30-38, diversity is associated with performance in dense-level tasks (e.g., detection and segmentation) that emphasize locality, while discrimination is linked to image-level classification tasks that require global separability. **Additional evidence supporting this relationship can be found in previous works such as [3][4]**. Specifically, [3] demonstrates that MIM with a locality inductive bias (diversity) outperforms in geometric and motion-related tasks, while CL with a more global attention focus (discrimination) excels in classification tasks, especially when the domain gap between pre-training and downstream is small. Additionally, [4] shows that MIM performs well in fine-tuning and dense prediction tasks, while the discrimination property of CL helps Vision Transformer models linearly separate images in their representation spaces. Therefore, combining the advantages of both diversity and discrimination proves to be effective across various types of downstream tasks.
>
> **Q3: [The notations need further clarification. ]:**
>
> In Figure 2(a), we examine the influence of various decoder configurations, including the scenario where no decoders are added.  We will improve our expression in the revised version.
>
> **Q4: [The claim that the increase in diversity derives from the asymmetric designs. ]:**
>
> As illustrated in Lines 50-51, **this claim mainly considers the scene of feature distillation, rather than discussing the characteristics of MIM pre-training**. Some previous works [1][2] have indicated that direct feature distillation can introduce feature diversity similar to the MIM mode, while we re-inspect in Section 2.2 where the diversity produced after distillation comes from. As for the MIM pre-training, as illustrated in Lines 137-139, its diversity derives from the pre-training mode no matter what the designs are and has no relation with the distillation scenario as discussed in Section 2.2.
>
> **Q5: [ Is there also a supervised training objective like cross-entropy loss?  ]:**
>
> To our knowledge, there is no supervised training objective like cross-entropy loss for distillation. The reason could be that there is no need to add such an objective for distillation. The student is solely trained by the distillation objective, and its discrimination ability is derived from the knowledge transferred by the supervised or CL pre-trained teachers. Similar to previous works [1][2], the model obtained by distillation needs further fine-tuning for downstream adaptation. Therefore, the inclusion of a supervised training objective in the distillation process is unnecessary.
>
> **Q6: [ How did the authors get the results of baseline methods? ]:**
>
> We obtained the CLIP classification result (83.6% for ViT-B) by fine-tuning it with a learning rate adjustment. The result already surpasses the fine-tuning results of [2] (82.9 for ViT-B) and the initial results of [17] (82.3 for ViT-B). [17] incorporates additional techniques such as EMA and LLRD to further boost the fine-tuning performance (achieving 85.7% for ViT-B). Although these tricks can de facto lead to performance gains, our study focuses on revealing the inherent characteristics of different distillation strategies and studying their effects on fine-tuning performance. Thus, we adopt the basic fine-tuning setting without considering these tricks for specifically yielding a high performance.
>
> **Q7: [ The gain of Hybrid Distillation over Distill-CLIP is not so significant ]:**
>
> Increasing diversity is mainly helpful for some dense-level tasks (e.g., detection and segmentation. We have discussed this issue in response iii) for Q2) that emphasize locality. As shown in Table 1 of our paper, our Hybrid Distill can greatly improve the performance for COCO detection and ADE20K segmentation compared to Distill-CLIP, i.e., achieving 1.1, 0.9, and 1.2 gains on $AP_{box}$, $AP_{mask}$ and mIoU, respectively. In addition, the increase in diversity does not cause discriminative loss, which is reflected in the performance of IN-1K compared to Distill-CLIP. Although the improvement is not as significant as using the DeiT teacher, the gains over the stronger CLIP baseline, especially for the dense-level tasks, still reflect the effectiveness of our Hybrid Distill.

---

> > ### Author Response · Authors · 2023-08-21
> >
> > Dear reviewer yZYM,
> >
> > Thanks again for all of your time on our work, and your suggestions have helped us a lot in improving the quality and clarity of this paper.
> >
> > We have further elaborated on the details and definitions that you have doubts about and highlighted the benefits of Hybrid Distill in dense-level tasks in the above rebuttal. Since the discussion deadline is approaching, we want to know whether our rebuttal has resolved your concerns and questions. We would appreciate it if you could check our response and provide feedback.
> >
> > Thanks very much for your effort!
> >
> > Best regards,
> >
> > Authors

---

### Official Review · Reviewer_dWYQ · 2023-07-05

**Soundness:** 1 poor
**Presentation:** 1 poor
**Contribution:** 2 fair
**Rating:** 5
**Confidence:** 5

**Summary:**

This work presents Hybrid Distillation, which attempts to distill from both supervised/CL and MIM frameworks. The work begins by revealing certain observations regarding the interplay between self-supervised pre-training and the concepts of diversity and discrimination. Subsequently, the authors propose the Hybrid Distillation technique that leverages token relations from the MIM teacher and feature maps from the supervised/CL teacher for knowledge distillation purposes.

**Strengths:**

The findings on the relationship between diversity and architecture are interesting

**Weaknesses:**

1. The presentation quality of this work is bad due to the following reasons:

i) Section 2 lacks an explanation of the experimental setup, For example, what is the design and architecture of the distillation? How are different decoders used for DeiT distillation?

ii) The description of the metrics, such as the average head distance and normalized mutual information, is inadequate. There is insufficient clarification regarding the existence of multiple attention distances for each layer and how they reflect diversity. Furthermore, the explanation of how NMI reflects discrimination is absent.

iii) The analysis of the figures lacks details, making it challenging to comprehend the meaning conveyed by the illustrated figures.

2. The authors state that "Mask Reconstruction of High-Level Semantics Does not Help Improve Diversity." However, previous studies [1][2][3][4] have distilled knowledge from high-level semantics and achieved high performance. Does this imply that high performance can still be attained with low diversity? If so, why should we care about diversity?

3. The correlation between the observations made and the design of the proposed Hybrid Distillation method is weak. For instance, how does the discovery of the "Asymmetric Decoder" inspire the proposed Hybrid Distillation approach?

4. The absence of a discussion and comparison of relevant works is noticeable. Numerous related works, such as [5][6][7], should have been included and compared.

5. Unfair comparisons are made in this work. While the proposed approach is distilled from multiple networks, it is only compared with methods that distill knowledge from a single network. Strong baselines that employ distillation from multiple networks should also be incorporated for a fair evaluation.


[1] IBOT : IMAGE BERT PRE-TRAINING WITH ONLINE TOKENIZER

[2] DINOv2: Learning Robust Visual Features without Supervision

[3] Masked Feature Prediction for Self-Supervised Visual Pre-Training

[4] BEiT v2: Masked Image Modeling with Vector-Quantized Visual Tokenizers

[5] Mimco: Masked image modeling pre-training with contrastive teacher.

[6] Layer Grafted Pre-training: Bridging Contrastive Learning And Masked Image Modeling For Label-Efficient Representations

[7] Contrastive Masked Autoencoders are Stronger Vision Learners

**Questions:**

See above

---

> ### Author Rebuttal · Authors · 2023-08-10
>
> **Q1: [The experimental setup in Section 2.]:**
>
> Please refer to the response to Q1 of Reviewer yZYM.
>
> **Q2: [The description of the metrics.]:**
>
> Please refer to the response to Q2 of Reviewer yZYM.
>
> **Q3: [The analysis of the figures lacks details.]:**
>
> The visualization of average head distance primarily provides insights into the nature of attention, distinguishing between long-range and short-range attention. The NMI visualization helps evaluate the relations between query tokens and key tokens. Low NMI indicates that attention maps are less dependent on the query tokens, suggesting that all queries attend to almost the same keys. We will add the details in the figure captions in the revised version.
>
> **Q4: [Why should we care about diversity?]:**
>
> i) Please refer to the response to Q2 iii) of Reviewer yZYM, diversity plays a crucial role in dense-level tasks.  Evidence can also be found in the results presented in Table 1, Table 2, and Table A1 of our paper.
>
> ii) As to the paper you mentioned, iBOT [11] and MaskFeat [12] employ an EMA teacher and HOG features, respectively, as the distillation targets. These works are covered in variants of our baseline work [1] and they demonstrate inferior performance compared to our Hybrid Distill across various downstream tasks. BEIT V2 [10] introduces an additional tokenizer training stage to refine the CLIP features and uses the refined features (codebook) as learning targets. DINOv2 [13] combines different techniques to scale the pretraining in terms of data (142M curated data) and model size (1B parameters). It then distills the 1B model into a series of smaller models, which differs from our Hybrid Distill in both motivation and distillation scales. The low-diversity problem may still exist in these works [10][13] but may be covered up by their additional design and large-scale training. Evidently, through the analysis of discrimination and diversity, our Hybrid Distill can achieve comparable results compared to [10] without introducing the tokenizer training stage and other distillation designs proposed in [10].
>
> **Q5: [The correlation between the observations made and the design.]:**
>
> i) The observation we made is to demonstrate that the increase in diversity of the previous distillation schemes [1][2] is obtained through the use of asymmetric designs, while it is challenging to introduce diversity without incorporating these designs (Section 2.2 and Section 2.4). Besides, we prove that the diversity brought by the asymmetric decoder is at the expense of sacrificing discrimination (Section 2.3). Accordingly, we argue that **single-model distillation cannot simultaneously ensure both diversity and discrimination, and this inspires us to further propose our Hybrid Distill strategy**.
>
> ii) Moreover, **many design details of our Hybrid Distill approach are directly influenced by the observations we made**. Specifically, we remove the asymmetric designs to promote discrimination in the student model (Lines 145-150). We also choose to distill token relations from the MAE teachers at layers where attention collapse tends to occur (Lines 168-173).
>
> **Q6: [The absence of a discussion and comparison of relevant works.]:**
>
> We will incorporate these relevant works [14][15][16] and include the following comparison:
>
> i) In [14], a momentum-updated target encoder is introduced as the contrastive learning objective and both reconstruction loss and contrastive loss are utilized. [15] further employ a pre-trained MoCov3 model as the target encoder and combine the pixel-level reconstruction loss with the contrastive loss. In contrast to them, our Hybrid Distill method directly distills knowledge from off-the-shelf MAE and CL pre-trained models and carefully designs the distillation location and target based on the analysis of properties. Hybrid Distill outperforms both these methods by jointly exploiting the off-the-shelf teachers and designing the distillation modes.
>
> ii) [16] observe that MIM and CL perform well at lower and higher layers, respectively. They then divide the pretrained model into three stages and utilize reconstruction loss and contrastive loss to constrain the first two stages and the last stage, respectively. Contrary to [16], our Hybrid Distill chooses to distill the MAE relation at the two layers before the last layer to alleviate the attention collapse and distill the CL features at the last layer to maintain discrimination. It is worth noting that we also explored a scheme similar to [16], where we divided the model into four stages at the L/4-th, L/2-th, and 3L/4-th layers (the 3rd, 6th, and 9th layers for ViT-B) and distilling the MAE relations at the first three stages. However, the results (49.1 $AP_{box}$) obtained using this scheme were inferior to our final schemes (50.0 $AP_{box}$).
>
> **Q7: [Compared with baselines that employ distillation from multiple networks.]:**
>
> We have evaluated our method with distillation from multiple networks in Table 5 in the paper. Specifically, we have reported the baseline that simultaneously distills the last layer features of both the MAE and DeiT teachers in the first row of Table 5. We also list these results and add another experiment on COCO that uses the MAE and CLIP teachers for 100 epochs distillation in the table below. It can be seen that directly using two teachers for feature distillation does not bring good results, especially when the weaker DeiT teacher is used. While by carefully designing the distillation mode, our Hybrid Distill can better utilize the respective advantages of the two teachers and bring consistent gains. **These results demonstrate that the performance gains are not merely from more teachers.**
>
> Method | Teachers  |$AP_{box}$| $AP_{mask}$|
> :----- | :----------: | :------: |------:
> Feature Distill | MAE and DeiT| 47.7|42.1 |
> Hybrid Distill | MAE and DeiT |50.0|43.9 |
> Feature Distill |MAE and CLIP|49.7|43.7 |
> Hybrid Distill |MAE and CLIP |50.4|44.1 |

---

> > ### Author Response · Authors · 2023-08-21
> >
> > Dear reviewer dWYQ,
> >
> > Thanks again for all of your constructive comments and suggestions, which have helped us improve the quality and clarity of this paper!
> >
> > We sincerely hope that our explanations and analyses could address your concerns and help you better understand the experimental details. We have also provided additional experiments in response to Q6 and Q7, which further demonstrated the effectiveness of our work. Since the deadline for discussion is approaching, we would like to kindly ask whether there are any additional concerns or questions that we might be able to address.
> >
> > Thanks very much for your effort!
> >
> > Best regards,
> >
> > Authors

---

### Official Review · Reviewer_CGRR · 2023-07-05

**Soundness:** 2 fair
**Presentation:** 2 fair
**Contribution:** 2 fair
**Rating:** 4
**Confidence:** 3

**Summary:**

This paper introduce a new distillation method that complimentary harmonizes two distillation method of different properties.

**Strengths:**

* Hybrid Distillation obtained higher accuracies than DeiT, MAE, and CLIP using them.
* Explanation with analyses (NMI, AHD, and attention map visualization)
* The paper is clearly written

**Weaknesses:**

* When I read the explanation in the method section, the proposed method seems very inefficient compared to methods without distillation (e.g., MAE, DINO). It would be better to compare quantitatively with throughput and total training hours.

* Some values in tables are different to original values in references.
    * The COCO detection performance of MAE in this paper and MAE paper are different. The AP^box and AP^mask of MAE are reported as 50.3% and 44.9% in MAE paper while they are reported as 48.4% and 42.6% in this paper, respectively.
    * The transfer learning result of MAE toward Naturalist19 is also different to the value in MAE paper. MAE paper report it as 80.5% for ViT-B while this paper report it as 75.2%.

* Are the explanations provided in the preliminary section the author’s own contributions or are they similar analyses  conducted and explained in other references? If they are the author’s contributions, which points can be regarded as novel?

* Some points are not understandable
    * The authors distilled the features of the supervised and CL models and MIM models to features of the different layers in student model regarding diversity and discriminatively. However, in the current design, the distillation performed on the last layer affect the front layers without detaching back-propagation of the distillation loss. Is this the intended situation?


**Questions:**

* How much more inefficient is Hybrid Distill compared to MAE?
* It may be better to remove unnecessary area in the right graph in Fig. 3(a)


**Limitations:**

* Its computational inefficiency was my primary concern, and the authors addressed it in the limitations section.

---

> ### Author Rebuttal · Authors · 2023-08-10
>
> **Q1: [The efficiency compared to methods without distillation.]:**
>
> i) Our Hybrid Distill, similar to many other distillation methods [1][2][8][9][10], requires an additional distillation stage after obtaining the pretrained model.  **The additional costs are inherent to distillation-based methods for they introduce additional distillation epochs**. Our Hybrid Distill requires 28 additional distillation hours with a batch size of 2048 for 300 epochs of distillation using ViT-B. However, for inference, the cost in terms of throughput (280 Imgs/s, measured on one 32G-V100 with a batch size of 64 and input size of $224^2$) remains the same since we do not modify the network structure.
>
> ii）In fact, since there is a growing availability of off-the-shelf pretrained models in the community, such as hugging face, it has become a more promising trend to consider how to utilize these pre-trained models efficiently, and **distillation is an efficient strategy for reusing these off-the-shelf model with better representation ability**.  Therefore, when directly exploiting these off-the-shelf models, it is not necessary to directly compare the training cost of pretraining (e.g., MAE and DINO) and distillation for a pretrained model.
>
> iii)  It would be more suitable to compare the training efficiency of our Hybrid Distill with the methods that directly perform single-model feature distillation, such as dBOT [1], and we have discussed it in Lines 71-77 of our appendix. Specifically, we have found that the per epoch training time of our Hybrid Distill is approximately 1.2x longer than that of using a single teacher, but our Hybrid Distill can achieve comparable or even better performance with fewer epochs, as indicated in Table A2 of our appendix. As a result, the total training time required for our Hybrid Distill only amounts to 22.5% of the training time required by dBOT.
>
>
> **Q2: [Some values in tables are different from the original values in references.]:**
>
> The difference between reported performance values and original values in the MAE paper is attributed to variations in downstream settings. We follow the settings of recent works (which are also widely adopted currently) for employing pretrained models in downstream tasks of COCO detection [1][5][6][7] and iNAT19 evaluation [1]. Experimental results demonstrate the consistent gains of our method compared to MAE. We further explain our settings and results below.
>
> i)  Regarding COCO detection, the MAE paper finetunes the pretrained model for 100 epochs, which is not a common downstream practice. Besides, MAE does not release the codes for COCO detection and some fine-tuning details are missing. Therefore, we choose to follow the common practice of previous works [5][6][7] that employs a 1x schedule (12 epochs) for fine-tuning, and our reported MAE results align with these papers (48.4 $AP_{box}$ and 42.6 $AP_{mask}$).
>
> ii) For the iNAT19 evaluation, the MAE paper also does not provide detailed information on the learning rate and training epoch settings. So we follow the settings of the baseline method dBOT [1] and retest the performance of MAE using their released model. Besides, all results in Table 2 of our paper, including iNAT19, are tested by ourselves in the same setting to ensure fair comparisons.
>
> iii) In Table A2 of our appendix, we also provide another set of COCO detection results using Cascaded Mask-RCNN. According to [1], the reported MAE result under this setting is 50.6 $AP_{box}$, while our Hybrid Distill achieves 53.0 $AP_{box}$, which is 2.4 points higher than MAE. This clearly demonstrates the advantages of our method compared to MAE across different settings.
>
> **Q3: [The explanations provided in the preliminary section.]:**
>
> In Section 2, our main contribution lies in the novel empirical findings that **shed light on the limitations of previous works [3][4] concerning both diversity and discrimination**. These findings are neglected in existing works, and consequently, discrimination and diversity cannot be guaranteed in a harmonized manner. In Section 2, we validate that the claims made in previous works, which suggest that feature distillation can enhance diversity, are not accurate. Specifically, it is difficult to increase diversity without incorporating asymmetric decoders (as shown in Sections 2.2 and 2.4), while these asymmetric decoders will sacrifice discrimination (Section 2.3). These findings are crucial for proposing our Hybrid Distill strategy in Section 3 which offers a solution to simultaneously ensure discrimination and diversity. Section 2.1 (the Preliminary section) mainly introduces the evaluation metrics we used for assessing the properties of the transformer models obtained by variant distillation settings. These metrics are not our novel contributions.
>
> **Q4: [The distillation performed on the last layer affects the front layers]:**
>
> i) Our intention is not to confine discrimination and diversity to a specific layer but rather to enhance the overall representation of the model by incorporating these properties.
>
> ii) In response to your suggestion, we attempt to detach the back-propagation of the CL feature distillation loss, restricting its impact solely to the last layer. However, we observe a significant decrease in performance using this scheme. We contend that constraining the gradient exclusively within the last layer results in substantial differences in feature distribution compared to other layers. Consequently, the model struggles to learn globally optimal representations, impeding its overall performance.
>
> Teacher | Detach  |$AP_{box}$| $AP_{mask}$|
> :----- | :----------: | :------: |------:
>  MAE and DeiT |Yes | 48.8|42.8 |
>  MAE and DeiT | No |50.0|43.9 |
> MAE and CLIP | Yes |49.0|43.0 |
> MAE and CLIP | No |50.4|44.1 |

---

> > ### Author Response · Authors · 2023-08-21
> >
> > Dear Reviewer CGRR,
> >
> > We would like to thank you again for your time and effort in reviewing our manuscript.
> >
> > It would be greatly appreciated if you could check our responses and provide your valuable feedback. We have given a more detailed explanation about your concerns, e.g. the efficiency and the value differences. We have also provided additional experiments that detach the back-propagation of the CL feature distillation loss and further demonstrated the effectiveness of our work.
> >
> > Since the deadline for discussion is approaching, please feel free to let us know if there are any additional clarifications or experiments that we can offer.
> >
> > Best regards,
> >
> > Authors

---

### Official Review · Reviewer_5Rvs · 2023-07-19

**Soundness:** 3 good
**Presentation:** 3 good
**Contribution:** 2 fair
**Rating:** 5
**Confidence:** 5

**Summary:**

The paper conducts sufficient experiments and theoretical analysis on diversity and discrimination.
Meanwhile,  the authors propose a simple yet effective hybrid distillation that combines contrastive learning pre-train and MIM pre-train.
This hybrid distillation achieves significant improvement on downstream tasks.


**Strengths:**

- The paper is well written, with sufficient experiments and analysis.
- The accuracy improvement is significant.

**Weaknesses:**

- The paper has some minor typo errors.


**Questions:**

1. Figure 4 seems to be wrong, it should be NMI(normalized mutual information), not attention distance.
2. For ViT-B, is the 300-epoch distillation saturated? Would more epochs of distillation bring larger gains?
3. For ViT-L, is there an obvious difference between using ImageNet-21K dataset and ImageNet-1K dataset? Is the performance improvement due to using ImageNet-21K dataset? Can you provide evaluation results trained on the ImageNet-1K dataset?
4. For the objective function, is it feasible to adjust the weight of the CL branch [e.g.$ \alpha D ( T_{c}(x) \odot M, S_{\theta}(M \odot x) ) $]?  Does the impact of branch CL have to be larger than branch MIM?
5. Are the classification results fair(CIFAR-100, Cars, INaturalist19 )? Can you use a unified setting to finetune the models(MAE, DeiT)?

**Limitations:**

- Hybrid Distill does not improve CLIP as much as DeiT after introducing the MAE teacher. The authors could spend more pages analyzing the relationship between the MIM branch and the CL branch, as well as the underlying reasons.

---

> ### Author Rebuttal · Authors · 2023-08-10
>
> **Q1: [Figure 4 seems to be wrong.]:**
>
> Figure 4 visualizes the per-layer average attention distance. In Figure 4, we further average the per-head attention distance and obtain a single average attention distance for each layer to reflect the overall distance in each layer and facilitate comparison between different models. As a result, the visualization may appear different from other per-head attention distance representations. We will provide additional explanations in the revised version.
>
>
> **Q2: [More distillation epochs.]:**
>
> As shown in the tables below,  **increasing the number of distillation epochs can enhance the performance, but the gain tends to be saturated and is relatively small.**  Since the purpose of our experiment is to verify the feasibility of hybrid distillation, for efficiency we did not prolong the training epochs to pursue further performance gains.
>
> Results for MAE+DeiT teachers
>
> Teacher | Epoch  |$AP_{box}$| $AP_{mask}$
> :----- | :----------: | :------: |------:
>  MAE and DeiT | 100 |50.0|43.9
>  MAE and DeiT | 300 |50.3|44.2
>  MAE and DeiT | 500 |50.4|44.2
>
> Results for MAE+CLIP teachers
>
> Teacher | Epoch  |$AP_{box}$| $AP_{mask}$
> :----- | :----------: | :------: |------:
>  MAE and CLIP | 100 |50.4|44.1
>  MAE and CLIP | 300 |50.6|44.4
>  MAE and CLIP | 500 |50.7|44.5
>
>
> **Q3: [Evaluation results trained on the ImageNet-1K dataset for ViT-L.]:**
>
>
> i) For ViT-L, we choose the Imagenet-21k pretrained MAE model instead of Imagenet-1k to ensure that the representation power of the MAE teacher is comparable to the large-scale CL pretrained CLIP model as much as possible and avoid one teacher dominating the other. Consequently, we continue to employ the ImageNet-21K data for the distillation process.
>
> ii) Following your suggestion, we also attempt to use the ImageNet-1K dataset for 300 epochs of distillation (the distillation epochs were set to 40 when using the ImageNet-21K dataset, ensuring comparable iterations ). As shown in the table below, the ImageNet-1K distillation results are very close to the ImageNet-21K distillation results, proving that the effectiveness of our Hybrid does not rely on the ImageNet-21K dataset. We will add these results in the revised version.
>
> Results for MAE+CLIP teachers
>
> Method | Dataset | Epoch  |IN-1K Acc.|COCO $AP_{box}$|COCO $AP_{mask}$| ADE20K mIoU
> :----- | :----------:| :------: | :------: | :------: | :------: |------:
> MAE|ImageNet-1K|-|85.9|54.0|47.1|53.6
> CLIP|CLIP-400M|-|86.1|52.7|46.2|54.2
> Hybrid Distill| ImageNet-21K| 40 |88.0| 54.6|47.6|56.3
> Hybrid Distill| ImageNet-1K| 300|87.6| 54.4|47.4|55.9
>
>
>
> **Q4: [Adjusting the weight of the CL branch.]:**
>
> Thanks for your suggestion. We have adjusted the loss weight of the CL branch ($\beta$), and the corresponding results on COCO are presented in the following tables. The first table pertains to MAE+DeiT teachers, while the second table relates to MAE+CLIP teachers. Compared to Table A.4 in our appendix, adjusting $\beta$ leads to a larger performance drop when using the CLIP teacher, as CLIP (pretrained with 400M image-text pairs) is more powerful than MAE (pretrained with 128K images).  While the impact on MAE+Deit teachers is relatively minor. In conjunction with Table A.4 in our appendix, we conclude that ensuring the weight of the more powerful teacher is necessary while setting both $\alpha$ and $\beta$ to 1 for simplicity, which can already yield satisfactory results.
>
> Results for MAE+DeiT teachers
>
> $\beta$ | 0  | 0.1 | 0.3 | 0.5| 0.7 | 1.0|
> :----- | :------:  | :------: |:------: |:------: |:------: |------:|
> $AP_{box}$|48.9|49.2 |49.7 |49.9 |50.0 |50.0|
> $AP_{mask}$|43.1|43.5|43.8 |43.9 |43.9 |43.9|
>
> Results for MAE+CLIP teachers
>
> $\beta$ | 0  | 0.1 | 0.3 | 0.5| 0.7 | 1.0|
> :----- | :------:  | :------: |:------: |:------: |:------: |------:|
> $AP_{box}$|48.9|49.3 |49.6 |49.9 |50.1 |50.4|
> $AP_{mask}$|43.1|43.5 |43.7 |43.9 |43.9 |44.1|
>
> **Q5: [Using a unified setting to finetune the models on CIFAR-100, Cars, INaturalist19.]:**
>
> We have already used a unified setting to finetune the MAE and DeiT models based on the dBOT [1] codebase. Specifically, the learning rate is set to 7.5e-6 for 512 batch size. the epochs are set to 1000 for CIFAR-100 and Cars and 360 for INaturalist19.

---

> > ### Comment · Reviewer_5Rvs · 2023-08-16
> >
> > Thanks for your reply, solved my problem, so I keep my score.

---

### Official Review · Reviewer_gLfn · 2023-07-25

**Soundness:** 4 excellent
**Presentation:** 3 good
**Contribution:** 3 good
**Rating:** 6
**Confidence:** 5

**Summary:**

This paper explores the subject of representation learning, focusing on two aspects: discrimination and diversity. Contrastive learning (CL) exhibits superior discrimination capabilities but suffers from limited diversity. Conversely, masked image modeling (MIM) offers greater diversity but shows weaker discrimination abilities.

The paper presents three insightful observations and integrates the benefits of both approaches, optimizing them for downstream tasks. In this context, "hybrid distillation" refers to the process where models are distilled using both contrastive learning (CL) for discrimination enhancement, and masked image modeling (MIM) for improving diversity. To reduce the training cost, the paper also proposes a token dropping strategy.

**Strengths:**

The paper has a solid experimental design to first re-examine discrimination and diversity, then propose its method and last to show its improvements on different downstream tasks. The idea is simple but effective and intuitive.

**Weaknesses:**

- Please ensure that the term "asymmetric X" is used consistently throughout the paper. The document refers to several variants of the term, including "asymmetric attention," "asymmetric architecture," "asymmetric decoder," and "asymmetric designs." It would be beneficial to differentiate between these concepts and clarify which is the primary focus of the paper.

- On line 104, the paper introduces the notation I() and H() to represent the mutual information and entropy, respectively, but does not explain how these quantities are calculated. For clarity, consider adding a brief explanation or citation for the methods used to estimate mutual information and entropy from the data.

- Similarly, on line 156 the notation S' is introduced without explanation. Please consider defining or explaining how S' is derived.

- When "feature distillation" is mentioned on line 106, adding a reference to the specific feature distillation approach used in the experiments would help clarify this concept for readers. Providing a citation would allow readers to refer to details of the feature distillation method.

**Questions:**

-

**Limitations:**

-

---

> ### Author Rebuttal · Authors · 2023-08-10
>
> **Q1: [Consistent usage of the terms "asymmetric X" throughout the paper]:**
>
>
> Thanks for the suggestions. The terms “asymmetric X” are consistent to indicate the asymmetric designs, which are our primary focus in Section 2. In detail, the terms '**asymmetric architecture**' and '**asymmetric designs**' share similar meanings, and in the revised version, we would use the unified term 'asymmetric designs' to encompass both concepts. The terms **asymmetric attention** and **asymmetric decoder** are detailed **asymmetric designs** that indicate the disparities of the attention layers in the encoder and the difference in the decoder side, respectively. We retain the two terms for clarification. In Section 2, we validate these 'asymmetric designs' and showcase their impact on diversity and discrimination. Through this validation, we aim to identify the limitations of previous works (Lines 42-47 of our paper) and lay the groundwork for introducing our Hybrid Distill method.
>
> **Q2: [How the notations I() and H() are calculated. ]:**
>
> The marginal entropy $H(q)$ and $H(k)$, as well as the mutual information $I(q,k)$, are computed using the following formulas:
>
> $H_{q}=-\sum_{q} p(q) \log p(q)$ ,
>
> $H_{k}=-\sum_{k} p(k) \log p(k)$ ,
>
> $I(q,k)=\sum_{q, k} p(q, k) \log \frac{p(q, k)}{p(q) p(k)}$.
>
> As explained in Lines 101-103, since we have $p(q)=\frac{1}{N}$ and $p(q, k)=\pi(k \mid q) p(q)$, we can determine the marginal distribution p(k) by summing over all the values of q, i.e., $p(k)=\sum_q p(q, k)$. With these distributions, we can subsequently calculate H(q), H(k), and I(q,k) accordingly. We have provided further clarification regarding the notations in the revised version.
>
> **Q3: [The notation S' is introduced without explanation ]:**
>
> $S_\theta^{\prime}$ represents the token relations of the student model in the layers preceding to the last layer and can be expressed similarly to $T_m^{\prime}$. Specifically, based on Equations (2) and (3) in the paper, $T_m^{\prime}(x)$ is defined as $\left[R_m^{L-1}(x), R_m^{L-2}(x)\right]$. The student model imitates the token relations in the same position as the teacher model. Therefore, $S_\theta^{\prime}=\left[R_s^{L-1}(x), R_s^{L-2}(x)\right]$, where $R_s^{i}(x)$ denotes the attention weight between queries and keys in the i-th layer of the student model.
>
>
> **Q4: [Adding a reference to the specific feature distillation approach.]:**
>
> Thanks for your suggestion and we will add the reference in the revised version. Specifically, the details of feature distillation align with [1][2], where only the features in the final layer of the ViT teacher are utilized as distillation objectives. We introduce various decoders, including linear projection and transformer decoders, in addition to the scenario without a decoder, to examine their impact on the feature representation. Subsequently, we compute the L1 loss between the decoded features of the student model and the features extracted from the teacher model.

---

### Author Rebuttal · Authors · 2023-08-10

We appreciate that the reviewers recognize the pros of our paper: the soundness (gLfn, 5Rvs, yZYM) and the contribution (gLfn, yZYM) of our paper, the experimental design to re-examine discrimination and diversity (gLfn, 5Rvs, CGRR, dWYQ), the simple but effective and intuitive idea (gLfn, yZYM), and the significant improvements on different downstream tasks (gLfn, 5Rvs, CGRR).

Although our presentation is affirmed by some reviewers (gLfn, 5Rvs, CGRR), we notice that some reviewers (dWYQ, yZYM) have expressed doubts about specific details in our paper. Common doubts (yBfq, Zj6k) lie in **the distillation settings, evaluation metrics, and our definitions of diversity and discrimination in Section 2**. We clarify that since one purpose of Section 2 is to verify the completeness of previous works [1][2], **the distillation settings are in line with these papers**. Similarly, the evaluation metrics we employed are commonly used for assessing the properties of transformer models [1][2][3][4]. Hence, we believe that our overall assessment process is robust. We also notice that some reviews have questions about **how we obtained the baseline results** (CGRR, yZYM). We clarify that **the baseline results are obtained by following the fine-tuning settings of previous works [1][2][5][6][7]**, and our Hybrid Distill results are also obtained under the same fine-tuning settings for fair comparisons. More details about these questions can be found in the point-by-point answers. We have also addressed the reviewers' other concerns individually and polished our paper to make it more comprehensible.

**Reference, also used for the individual response:**

[1] Exploring target representations for masked autoencoders, 2022.

[2] Contrastive Learning Rivals Masked Image Modeling in Fine-tuning via Feature Distillation, 2022.

[3] Revealing the Dark Secrets of Masked Image Modeling, 2022.

[4] What Do Self-Supervised Vision Transformers Learn? 2023.

[5] Context Autoencoder for Self-Supervised Representation Learning, 2022.

[6] Sdae: Self-distillated Masked Autoencoder, 2022.

[7] Progressively Compressed Auto-Encoder for Self-Supervised Representation Learning, 2023.

[8] Eva: Exploring the limits of masked visual representation learning at scale, 2022.

[9] Mvp: Multimodality guided visual pre-training. 2022.

[10] BEiT v2: Masked image modeling with vector-quantized visual tokenizers, 2022.

[11] IBOT: IMAGE BERT PRE-TRAINING WITH ONLINE TOKENIZER, 2021.

[12] Masked Feature Prediction for Self-Supervised Visual Pre-Training, 2021.

[13] DINOv2: Learning Robust Visual Features without Supervision, 2023.

[14] Contrastive Masked Autoencoders are Stronger Vision Learners, 2022.

[15] Mimco: Masked Image Modeling Pre-training with Contrastive Teacher, 2022.

[16] Layer Grafted Pre-training: Bridging Contrastive Learning And Masked Image Modeling For Label-Efficient Representations, 2023.

[17] CLIP Itself is a Strong Fine-tuner: Achieving 85.7% and 88.0% Top-1 Accuracy with ViT-B and ViT-L on ImageNet, 2022.

---

### Decision · Program_Chairs · 2023-09-21

**Decision:**

Reject

**Comment:**

This paper proposes a hybrid distillation method that combines contrastive learning and masked models. Extensive experiments are conducted to show this method archives SotA results.  All reviewers respect that the paper gets SoTA results on self-supervised representation with the proposed hybrid distillation.  Concerns include (a) writing needs to be polished (gLfn, dWYQ, yZYM), and should include more technique details and results compared with strong baselines (yZYM, dWYQ).

The meta-reviewer has read all reviews, the paper, and the rebuttal. The paper's experimental results are valuable, to some extent, to the self-supervised learning. However, the meta-reviewer agrees with the reviews that the clarity of the writing and the technical details need to be polished before submitting it to the next venue.